# Whey- and Soy Protein Isolates Added to a Carrot-Tomato Juice Alter Carotenoid Bioavailability in Healthy Adults

**DOI:** 10.3390/antiox10111748

**Published:** 2021-10-31

**Authors:** Mohammed Iddir, Denis Pittois, Cédric Guignard, Bernard Weber, Manon Gantenbein, Yvan Larondelle, Torsten Bohn

**Affiliations:** 1Nutrition and Health Group, Department of Population Health, Luxembourg Institute of Health, 1445 Strassen, Luxembourg; mohammed.iddir@lih.lu; 2Louvain Institute of Biomolecular Science and Technology, UCLouvain, 1348 Louvain-la-Neuve, Belgium; yvan.larondelle@uclouvain.be; 3Department Environmental Research & Innovation (ERIN), Luxembourg Institute of Science and Technology, 4422 Belvaux, Luxembourg; denis.pittois@list.lu (D.P.); cedric.guignard@list.lu (C.G.); 4Laboratoires Réunis Luxembourg S.A., 6131 Junglinster, Luxembourg; bernard.weber@labo.lu; 5Luxembourg Institute of Health, Translational Medicine Operations Hub, Clinical and Epidemiological Investigation Centre, 1445 Strassen, Luxembourg; manon.gantenbein@lih.lu

**Keywords:** carotenoids, enzymes, micellization, plant vs. animal proteins, emulsification, xanthophylls

## Abstract

Recent findings suggested that proteins can differentially affect carotenoid bioaccessibility during gastro-intestinal digestion. In this crossover, randomized human trial, we aimed to confirm that proteins, specifically whey- and soy-protein isolates (WPI/SPI) impact postprandial carotenoid bioavailability. Healthy adults (*n* = 12 males, *n* = 12 females) were recruited. After 2-week washout periods, 350 g of a tomato-carrot juice mixture was served in the absence/presence of WPI or SPI (50% of the recommended dietary allowance, RDA ≈ 60 g/d). Absorption kinetics of carotenoids and triacylglycerols (TAGs) were evaluated via the triacylglycerol-rich lipoprotein (TRL) fraction response, at timed intervals up to 10 h after test meal intake, on three occasions separated by 1 week. Maximum TRL-carotenoid concentration (C_max_) and corresponding time (T_max_) were also determined. Considering both genders and carotenoids/TAGs combined, the estimated area under the curve (AUC) for WPI increased by 45% vs. the control (*p* = 0.018), to 92.0 ± 1.7 nmol × h/L and by 57% vs. SPI (*p* = 0.006). Test meal effect was significant in males (*p* = 0.036), but not in females (*p* = 0.189). In males, significant differences were found for phytoene (*p* = 0.026), phytofluene (*p* = 0.004), α-carotene (*p* = 0.034), and β-carotene (*p* = 0.031). C_max_ for total carotenoids (nmol/L ± SD) was positively influenced by WPI (135.4 ± 38.0), while significantly lowered by SPI (89.6 ± 17.3 nmol/L) vs. the control (119.6 ± 30.9, *p* < 0.001). T_max_ did not change. The results suggest that a well-digestible protein could enhance carotenoid bioavailability, whereas the less digestible SPI results in negative effects. This is, to our knowledge, the first study finding effects of proteins on carotenoid absorption in humans.

## 1. Introduction

Carotenoids are lipid-soluble pigments produced by plants, bacteria, and certain fungi. These secondary plant compounds cannot be produced by the human body, but are present in the blood and tissues as they are taken up from a large variety of fruits and vegetables. These pigments have been associated in epidemiological studies with the reduced incidence of several chronic diseases [1], such as lowered risk of coronary heart diseases and type 2 diabetes [2], certain types of cancer [3], as well as age-related macular degeneration [4]. The possible role of carotenoids in disease prevention is far from fully understood. However, this may be attributed to their anti-inflammatory properties, but also to their antioxidant functions, where carotenoids are mostly involved in the scavenging of reactive oxygen species (ROS) such as singlet oxygen or peroxyl radicals, thanks to the number of conjugated double bonds present in the molecule, which determines to a large extent their antioxidant properties [5,6,7]. Carotenoids have also been shown to exert an influence on the cellular level, by acting on gene transcription [8]. Beyond these properties, certain carotenoids act as precursors of vitamin A and constitute a dietary source for many subjects not consuming sufficient animal-based foods, notably vegetarians/vegans, but also for many living in developing countries [9,10].

These aspects stimulated interest in carotenoid absorption and metabolism, though the bioavailability of carotenoids is low and variable due to the multiple factors that affect their release, absorption, transport, metabolism, and storage [11,12]. The type of carotenoid species [13,14], dietary aspects such as matrix, as well as host-related factors [15,16] are known to interfere with these above-mentioned metabolic steps. The type of carotenoid species may play a role as they differ in their polarity, which could arbitrate the extent of micellization, also known as bioaccessibility. It has also been acknowledged that the food matrix from which they are absorbed plays a role. For instance, lipids can enhance carotenoid absorption efficiency, by improving their micellization and fostering chylomicron sequestration [17,18], while dietary fibers could negatively affect their bioavailability, by hampering the transition of lipid droplets into mixed micelles and/or affecting the activity of digestion enzymes [19,20]. Additionally, it has been hypothesized that co-consumed dietary proteins, being highly surface-active, adsorb to the lipid droplets and may aid in emulsifying apolar dietary constituents by modulating their transfer into mixed micelles [21]. During digestion, proteins have also been reported to have the potential to act as antioxidants. This may be increased by exposing their amino acids by disrupting their tertiary structures, which can inhibit lipid oxidation through multiple pathways, including the inactivation of ROS, scavenging free radicals, chelation of pro-oxidative transition metals, reduction of hydroperoxides, and alterations of the physical properties of food systems [22]. In our previous in vitro studies, we have shown that various proteins (whey protein isolate (WPI), soy protein isolate (SPI), sodium caseinate (SC), and gelatin (GEL)) had positive and negative effects on the bioaccessibility of carotenoids originating from different matrices such as carrot juice, tomato juice, and spinach, depending on the type of carotenoids [23], but also on the type and concentration of proteins [14]. In fact, higher carotenoid polarity was associated with a stronger negative influence of proteins on their bioaccessibility, while hydrophobic carotenoids appeared in part to benefit from the presence of proteins regarding their micellization. As carotenoid micellization is assumed to be a key step for carotenoid bioavailability, we have suggested that if a similar interaction occurs in vivo, the presence of proteins may also influence carotenoid absorption efficiency.

In the present crossover randomized human study, we aimed to prove or disprove the hypothesis that proteins, and specifically WPI and SPI, can impair postprandial carotenoid bioavailability in healthy humans. For this purpose, a carotenoid-rich test meal, i.e., a mixture of tomato and carrot juice, was served in the absence or presence of proteins at a concentration equivalent to 50% of the recommended dietary allowance (RDA ≈ 60 g/d). Absorption kinetics of carotenoids were evaluated by measuring the triacylglycerol-rich lipoprotein (TRL) fraction response, representing the newly absorbed carotenoids, at timed intervals up to 10 h after test meal intake.

## 2. Materials and Methods

### 2.1. Chemicals, Carotenoid Standards and Food Matrices

All chemicals were of analytical grade or superior. Methanol (MeOH), acetonitrile (ACN), dichloromethane (DCM), diethyl ether (DEE), hexane (HEX) were purchased from Carl Roth GmbH + Co. KG (Rotisol^®^, Karlsruhe, Germany), while methyl tert-butyl ether (MTBE) and butylated hydroxytoluene (BHT) were purchased from Sigma-Aldrich (Merck KGaA, Darmstadt, Germany) and ammonium acetate purchased from Scientest Biokemix GmbH (Leese, Germany), ethanol, sodium chloride and anhydrous sodium sulfate from VWR (Leuven, Belgium). Ultrapure water (18 MΩ) was used throughout the study and was prepared with a purification system from Millipore (Brussels, Belgium).

Standards of all-trans-β-carotene (β-Car, powder, ≥97%, Art. No. 22040), all-trans-lycopene (Lyc, powder, ≥85%, Art. No. 75051), α-carotene (α-Car, powder, ≥97%, Art. No. 50887), neoxanthin (Neo, powder, ≥97%, Art. No. 72994), and trans-β-apo-8′-carotenal (internal standard (IS), powder, ≥96%, Art. No. 10810) were purchased from Sigma-Aldrich (St. Louis, MO, USA); lutein (Lut, powder, ≥95%, Art. No. 0306S), zeaxanthin (Zea, powder, ≥98%, Art. No. 0307S), and β-cryptoxanthin (β-Cry, powder, ≥97%, Art. No. 0317S) were obtained from Extrasynthese (Genay, France); violaxanthin (Vio, powder, ≥95%, Art. No. 0259), (9Z)-β-carotene ((9Z)-β-Car, powder, ≥95%, Art. No. 0003.1), phytoene (Pte, oily, ≥95%, Art. No. 0044), and phytofluene (Ptf, oily, ≥95%, Art. No. 0042) were acquired from Carotenature GmbH (Münsingen, Switzerland).

Tomato juice (Delhaize Le Lion, Belgium), organic carrot juice (Delhaize Le Lion, Belgium), Delhaize peanut oil (free of native carotenoids [24] and own blank examinations), butter (Carlsbourg, Luxembourg province, Belgium), cream cheese (La Vache qui rit, Suresnes, France), bread (Harrys, Boulogne-Billancourt, France), minced turkey fillet (Delhaize Le Lion, Belgium), Greek yogurt (Mevgal, Thessaloniki, Greece), apples (Golden, Delhaize Le Lion, Belgium), and water (SPA Monopole, Spa, Belgium) were purchased from a local supermarket (Delhaize, Strassen, Luxembourg).

Soy protein isolate (SPI) was obtained from Self Omninutrition^®^ (≥90% purity, Stockholm, Sweden), and whey protein isolate (WPI) was acquired from Pure Nutrition USA (≥90% purity, Oxnard, CA, USA).

### 2.2. Participants

Twenty-four individuals (twelve men and twelve women) from or living in the region of the Grand Duchy of Luxembourg were recruited for this study. Subjects were 20 to 50 years old, non-smokers (abstinent for at least 2 years), and body-mass index (BMI) < 30 kg/m^2^ (Table 1).

Participants had no history of metabolic disease, malabsorption disorders, hyperlipidemia, hyper-glycaemia, food allergies/intolerances/special diets that are not compatible with test meals or washout periods, and were not (1) consuming regularly dietary supplements during the study, (2) eating more than 5 portions of fruits and vegetables per day, (3) taking any medication for chronic conditions or recent illness (e.g., antibiotics), (4) practicing daily intense physical activity (≥120 min). Vegan/vegetarian subjects were likewise excluded. The number of participants was based on a randomized block design, consisting of six blocks of four subjects each, with each being served different sequences (A/B/C; A/C/B; B/A/C; B/C/A; C/A/B and C/B/A) of test meals (A = control; B = WPI; C = SPI) in order to cancel out any potential effect of sequence of meals (Figure 1). 

The National Research Ethics Committee of Luxembourg (CNER) approved the present study (n° 201710/04). Procedures followed were in accordance with the Declaration of Helsinki, and according to the ICH-GCP (international Conference on Harmonization-Good Clinical Practice) and with European General Data Protection Regulations (GDPR). The study has been registered under the reference number NCT04078646 on the website of clinicaltrials.gov (accessed on 27 September 2021).

### 2.3. Study Procedure

The main steps of the study included recruitment, an information session, a first screening visit, and the experimental parts.

#### 2.3.1. Recruitment

The study was carried out at the Clinical and Epidemiological Investigation Center (CIEC) of the Luxembourg Institute of Health (LIH). Candidates were informed about the study by resorting to flyers, local radio and newspapers, e-mails, social media, “word of mouth” communication, and advertisement at the LIH website. 

#### 2.3.2. Information Session

This appointment was meant to inform the participants about the trial, answer all questions that participants might have, and have participants sign the Informed Consent Form (ICF) and fill out our health and lifestyle questionnaire, which was used to determine the person’s eligibility for this study.

#### 2.3.3. Screening Visit

Spot urine and blood samples were collected from the participants, and were meant to check the subjects’ anemic status and analyze glucose levels as well as serum lipids such as triacylglycerols (TAGs) and cholesterol fractions.

#### 2.3.4. Experimental Design

Final eligible participants commenced the four-week trial phase, which included three washout periods, one short enrolment visit, and three full day clinical visits. 

The enrolment visit (also termed baseline visit), took place at the beginning of the trial phase. A blood sample was collected to determine the baseline levels of TAGs and plasma carotenoids at the beginning of the trial, prior to the first washout phase. 

The first washout period started on day 1 after the baseline visit and had a duration of two weeks during which the participants were asked to stay on a low carotenoid diet (i.e., to avoid the intake of colored fruits and vegetables), to reduce the basal levels of plasma carotenoids, before reporting to the CIEC for the 1st clinical visit (Figure 2). Depending on the test day, proteins (either SPI or a WPI) were or were not added to the breakfast meal. This first clinical full-day visit (detailed in the following paragraph) was followed by a third week of washout, during which the participant continued on a low carotenoid diet. At the end of the third week of washout, a second appointment at the CIEC took place. This was followed by the 4th week of washout period and at the end, the participant had the third and last appointment at the CIEC. 

A list of foods to be avoided during the washout period, together with a list of alternative food items were given to each participant. The volunteers were asked to fill in a provided food journal, on a daily basis, where they wrote down the portions of what they had eaten during the day. This was used to check compliance with the washout period and to better interpret later findings. 

#### 2.3.5. Clinical Visit and Postprandial Experiments

Recruitment procedures resulted in the final inclusion of 12 men and 12 women. On clinical visit days, participants were asked to report to the CIEC’s facilities, starting from 7:30 a.m., and a baseline blood sample (20 mL) was drawn (“0 h” time point). A carotenoid-rich test meal was then immediately served after the baseline blood draw (Table 2), and was composed of carrot and tomato juice mixture (1:1, *v/v*) as a rich source of carotenoids (Table 3), to which peanut oil was added. Depending on the test day, proteins (either SPI or a WPI) were or were not added to the carotenoid-rich meal. A toasted bread (white wheat) spread with butter and cream cheese, as well as water low in minerals were also served (Table 2). The entire test meal was consumed within 30 min, under supervision. 

Post-prandial blood samples (20 mL each) were collected at timed intervals (before breakfast meal intake, i.e., hours 0, 2, 3, 4, 5, 6, 8, and 10 after breakfast meal intake) by a trained nurse from the forearm into EDTA-K2 and SSTTM II Advance tubes (BD Vacutainer^®^, Fisher Scientific, Illkirch, France), for carotenoid and TAGs analysis, respectively. Participants received a standardized lunch 4 h after breakfast meal intake, consisting of a toasted sandwich (white wheat bread), with turkey, some butter to spread on the bread, Greek yogurt, and a small apple. The participants had a courtesy meal at the end of the visits for dinner, i.e., 10 h after breakfast meal intake. No other foods or beverages except water (ad libitum) were allowed during the day (including during breakfast and lunch if desired). The three breakfast meals were given to the participants in six different permutations (Figure 1).

### 2.4. Sample Processing

The collected blood samples for carotenoid analysis were immediately centrifuged at 2000× *g* for 8 min, at 4 °C (Sigma 2-16KC centrifuge, Thermo Fisher Scientific Inc., Waltham, MA, USA). The obtained blood plasma was used for the separation of the TRL fraction rich in chylomicrons, as described earlier [25,26]. Briefly, 2.5 mL of plasma was over-layered at room temperature with 2.2 mL of a 1.006 g/L NaCl solution in a 5 mL ultra-clear open-top tube (Beckman Coulter), and TRL fractions were separated at 180,000× *g* for 1 h at 24 °C in an Optima MAX-XP Ultracentrifuge (Beckman Coulter), using the high-capacity swinging-bucket MLS-50 rotor. Following the centrifugation, the bottom of the tubes was punctured with a heated needle to remove the lower plasma phase, and the upper TRL fraction (1–1.2 mL) was collected into 1.5 mL cryovials. The volume was brought up to 1.5 mL using 1.006 g/L NaCl by rinsing the ultra-clear tubes before storage at −80 °C. Likewise, plasma aliquots were stored at −80 °C for carotenoid analyses in order to measure the effect of the washout periods.

Fasting blood glucose, lipids, and hematology as well as serum-TAG analyses were performed by a local external commercial laboratory (Laboratoires Réunis, Junglinster, Luxembourg). The samples were centrifuged for 5 min at 3775× *g*, at 4 °C, and kept on site at 4 °C until pickup and analysis on the same day.

### 2.5. Extraction of Carotenoids

The extraction of carotenoids from tomato and carrot juice samples was performed as described earlier [14,27]. Blood plasma and plasma TRL fraction extraction protocols were adapted from Unlu et al. [26]. In short, 1.5 mL of TRL fraction or 1.0 mL of plasma were thawed and totally transferred into 15 mL Falcon tubes. A total 3.25 mL of ethanol containing 0.1% of butylated hydroxytoluene (BHT) were added to the samples, which were vortexed and spun down at 4 °C for 2 min, at 600× *g*, in order to collect the supernatant into a new tube. The precipitates were re-extracted with 3 mL of diethyl ether–hexane (1:2, *v/v*) containing 0.02% BHT, and centrifuged at 1250× *g* for 2 min, at 4 °C, and the organic upper phases were combined. A total 2 mL of saturated NaCl as well as 4 mL of hexane (plus 0.02% BHT) were added to the combined supernatants before their spinning down at 1250× *g* for 2 min, at 4 °C. Another extraction step was performed on the combined extracts with 3 mL of diethyl ether–hexane (1:2, *v/v* + 0.02% BHT). 

To ensure that the combined organic phases did not contain any traces of water, a small amount of sodium sulfate (water free, ≈500 mg) was added to each tube, and the organic phase was transferred into a new 15 mL tube and dried under a stream of nitrogen for ≈40 min at 30 °C (TurboVap LV from Biotage, Uppsala, Sweden). Tubes were then flushed with argon and stored at −80 °C until analysis.

### 2.6. Carotenoid Analysis

The analysis of carotenoids was carried out as described previously [14]. In brief, the dried extracts were re-dissolved with fridge-cold MTBE/MeOH (30/70, *v/v*) in a total volume of 150 μL (TRL fraction), 600 μL (plasma samples) or 6 mL (tomato and carrot juice extracts). 

The internal standard (IS, *trans*-β-apo-8′-carotenal) was added to each sample to reach a final concentration of 0.2 μg/mL. HPLC separation of carotenoids was carried out using an Agilent 1260 Infinity HPLC instrument (Agilent Technologies, Santa Clara, USA), equipped with an Accucore C30 reversed phase column (2.6 μm particle size, 100 mm length, 2.1 mm diameter, Thermo Fisher Scientific). The mobile phase consisted of water/MeOH (60/40, *v/v*) with 30 mM of ammonium acetate as eluent A and ACN:DCM (85/15, *v/v*) as eluent B. Elution gradient was as follows: 0 min 42% B; 4 min 48% B; 5 min 52% B; 11 min 52% B; 13 min 75% B; 18 min 90% B; 40 min 90% B; 41 min 42% B. The flow rate was fixed at 0.5 mL/min, the injection volume was 10 μL, and column temperature was 28 °C. Seven point-linear calibration curves were prepared with external standards for each compound, with concentrations ranging from 60 to 500 ng/mL. Peaks were integrated at 286 nm (Pte), 350 nm (Ptf), 440 nm (Neo and Vio), 450 nm (Zea, Lut, α-Car, β-Car, and (*9Z*)-β-Car), 455 nm (β-Cry and IS), and at 470 nm (Lyc). Carotenoids were detected with a diode array detector, and identified by comparing each carotenoid’s retention time and absorption maxima [28], with those of the available standards. Concentrations of carotenoids were determined using the internal standard method.

### 2.7. AUC, C_max_, T_max_

The postprandial AUC of time vs. concentration of the respective carotenoids extracted from TRL fractions was determined on the basis of seven postprandial time points plus the baseline (fasting state, before breakfast meal intake). The AUC was then determined from baseline-subtracted (each concentration was subtracted from its baseline value) values using the trapezoidal method. In case values below baseline were observed, these were included into the AUC calculations (as negative values) in order to maintain a maximum amount of information. Individual C_max_ values reflected the highest carotenoid concentrations measured in the TRL fraction of one participant on one clinical day, irrespective of the time point, and these were used to calculate the average C_max_. Individual T_max_ values of one person, at each individual visit, were also noted down.

### 2.8. Statistical Analyses

Normality of distribution of obtained AUC values as well as equality of variances were tested by Q-Q plots and box-plots, respectively. When needed, non-normally distributed data was log-transformed for further statistical analyses. A linear mixed model was created, with each subject acting as their own control, with AUC or C_max_ as the dependent, observed factor (main outcomes). Fixed factors were type of carotenoid or TAG (with levels of Lyc, β-Car, α-Car, Pte, Ptf, TAG), breakfast meal (A, B, or C), breakfast meal sequence, i.e., random block number (levels from 1–6), as well as the individual (ID number) nested within sequence and gender. Following significant Fisher-F tests for breakfast meal, Fisher protected LSD tests were used to compare the three treatments against one another, while Tukey’s post-hoc test was employed for studying all group-wise comparisons if number of groups was >3. Following significant interactions (e.g., between test meal and type of carotenoid), one factor was kept constant to re-investigate statistical findings. *p*-values below 0.05 (2-sided) were considered significant. Unless otherwise reported, all values are means ± SD. SPSS (vs. 25, IBM, Chicago, IL, USA) was used for all analyses.

## 3. Results

### 3.1. Description of Subjects

Thirty healthy and free-living participants were enrolled in the study. Among these, six dropped out of the study for personal reasons, and twenty-four participants completed the trial, i.e., twelve males and twelve females consumed all three breakfast meals, from which data were analyzed. Anthropometric measures and blood chemistry profile of the subjects at baseline are shown in Table 1. The baseline characteristics of the participants were in line with the inclusion criteria, but it is worth mentioning that one male participant had borderline TAG values (≈190 mg/dL), while two female participants had elevated total cholesterol concentrations (≈230 mg/dL). However, in no case both TAG and cholesterol were elevated, and the participants were advised to follow-up their blood lipids. No adverse reactions were observed during the three clinical days or in between.

### 3.2. Breakfast Meal and Carotenoid Composition

Apart from the breakfast meal A (control, no additional proteins (1281 kJ/355 kcal)), test meals B (added WPI) and C (added SPI) were isocaloric (1759 kJ/467 kcal and 1753 kJ/468 kcal, respectively) (Table 2). For all test meals, a total of 350 g of the juice were administered to the subjects for breakfast, being a 1/1 mix (*v/v*) of carrot and tomato juices. Carotenoid compositions and concentrations (mg/100 g) are reported in Table 3.

### 3.3. Plasma Levels and Effect of Washout

The participants’ fasting carotenoid plasma concentrations at baseline are reported in Table 4. Women had higher carotenoid plasma levels than men (by approx. 45%). The concentration of total plasma carotenoids was generally reduced by the washout period, from 455 (D_0_) to 305 (V_1_) and then to 285 nmol/L (V_2_ and V_3_) (Table 4). The decrease from D_0_ to V_1_ was significant (*p* < 0.001), while V_1_, V_2_, and V_3_ carotenoid plasma values did not differ significantly. This was also observed for each individual plasma carotenoid measured (except Lut+Zea and β-Cry for male). Although the effect of the two weeks of initial washout was significant in both genders, a stronger decrease was found in males compared to females (a 45% vs. 30% reduction, on average).

### 3.4. Effect of Proteins on Carotenoid AUC of Plasma-TRL Fraction over Time

#### 3.4.1. Total Population (*n* = 24)

Following linear mixed models, it was tested whether there was any significant effect of the given meals on the AUC of serum TAGs/plasma-TRL fraction of carotenoids. The effect of the employed meals on AUC (all carotenoids investigated considered plus TAGs, i.e., statistically pooled) was significant (*p* = 0.011), as was the effect of gender and type of carotenoid (both *p* < 0.001). Considering all three meals and all types of carotenoids combined, AUC for carotenoids (nmol × h/L ± SEM) and TAGs (mg × h/dL ± SEM) were higher in males, compared to females (99.0 ± 2.0 vs. 44.7 ± 1.9, *p* < 0.001, and 339.9 ± 7.8 vs. 196.7 ± 7.6, *p* = 0.01, respectively) (Table 5).

Considering both genders and all carotenoids/TAGs investigated (statistically pooled) combined, the overall AUC (nmol × h/L) for test meal B (WPI) increased by 45% compared to the meal A (control, *p* = 0.018), and by 57% compared to test meal C (SPI, *p* = 0.006), while the latter did not differ significantly, as compared to test meal A (*p* = 0.685).

AUC for total carotenoids (nmol × h/L ± SEM) was 186 ± 7.0, 249 ± 7.0 and 124 ± 7.0 for control, WPI and SPI meals, respectively, being significantly different between the WPI meal and SPI meal (*p* = 0.014, Figure 3). Similarly, significant differences were found between SPI and WPI for individual carotenoids, i.e., Pte (*p* = 0.002), Ptf (*p* = 0.001), and α-Car (*p* = 0.002), although the difference between WPI vs. control meal was also significant for Ptf (Table 5). AUC values for TAGs (mg × h/dL ± SEM) were significantly higher for the WPI meal (332.5 ± 9.6), than for the control meal (196.3 ± 9.6) (*p* = 0.044, Figure 3).

#### 3.4.2. Males (*n* = 12)

The effects of the employed meal and carotenoid were significant (*p* = 0.036 and *p* < 0.001). Again, the AUC (nmol × h/L) for WPI increased by 46% compared to the control meal (*p* = 0.045), and by 62% compared to SPI (*p* = 0.016) (all carotenoids/TAGs considered, i.e., statistically pooled).

AUC for total carotenoids (nmol × h/L ± SEM) was 279.2 ± 17.4 for the control meal, 364.5 ± 17.4 for WPI and 171 ± 17.4 for SPI, with a significant difference between the tested proteins (WPI vs. SPI, *p* = 0.034). Similarly, significant differences were found between WPI- and SPI-supplemented meals for individual carotenoids including Pte (*p* = 0.026), Ptf (*p* = 0.004), α-Car (*p* = 0.034), and β-Car (*p* = 0.031) (Table 5). However, the AUC for TAGs did not significantly differ between the three meals (Figure 3).

#### 3.4.3. Females (*n* =12)

The effect of carotenoid was significant (*p* < 0.001), while it was not significant for the test meal effect (*p* = 0.189). Regarding the carotenoid effect, AUC for total carotenoids did not differ significantly between the test meals, while for individual carotenoids, significant differences were found only for Pte, with significantly lower AUC values (nmol × h/L ± SEM) for the SPI meal (63.3 ± 11.0), as compared to the control meal (94.5 ± 14.9, *p* = 0.013), and the WPI meal (112.2 ± 21.7, *p* < 0.001) (Table 5). The AUC values for TAGs (mg × h/dL ± SEM) were significantly lower for the control meal vs. the WPI meal (165.3 ± 6.3 vs. 238.2 ±5.9, *p* = 0.024) (Figure 3).

### 3.5. Effect of Proteins on Maximum Plasma TRL-Carotenoid Concentration (C_max_)

#### 3.5.1. Total Population (*n* = 24)

The effects of tested meal and carotenoid were significant (both *p* < 0.001) regarding C_max_, while the effect of gender was not (*p* = 0.116). 

C_max_ for total carotenoids (nmol/L ± SD) was 119.6 ± 30.9 vs. 135.4 ± 38.0 vs. 89.6 ± 17.3 nmol/L for test meals A (control, no protein), B (WPI), and C (SPI), respectively, being significantly different between control meal and SPI, as well as between WPI and SPI (both *p* < 0.001, Figure 4B). Similarly, the C_max_ of several individual carotenoids such as Lut+Zea, Lyc, Pte, α-, and β-Car were significantly lower in the presence of SPI, as compared to the WPI and control meals (*p* < 0.05, Figure 4(A3)). However, the C_max_ values for TAGs (mg/dL ± SD) were only significantly different between the control meal and WPI (143.3 ± 41.4 vs. 162.4 ± 49.2 mg/dL, *p* = 0.044, Figure 4C).

#### 3.5.2. Males (*n* = 12)

The effects of test meal and carotenoid type on C_max_ were significant (both *p* < 0.001). C_max_ for total carotenoids (nmol/L ± SD) was significantly lower in the presence of SPI, compared to control meal and WPI-supplemented test meal, by approx. 33 and 36%, respectively (88.0 ± 21.1 vs. 131.6 ± 41.9 and 138.6 ± 50.0, respectively; both *p* < 0.001) (Figure 4B). Significant differences were found for C_max_ of several individual carotenoids, such as Pte, Ptf, α-Car, and β-Car, between the SPI-supplemented meal and the two other meals (*p* < 0.05, Figure 4(A1)), while the C_max_ values for TAGs (mg/dL ± SD) did not significantly differ between the three meals (Figure 4C).

#### 3.5.3. Females (*n* = 12)

The effects of tested meal and type of carotenoid were also both significant (*p* < 0.001). C_max_ for total carotenoids (nmol/L ± SD) was significantly different between the control meal (107,7 ± 12.8) and WPI (132.1 ± 23.0, *p* = 0.003), and between control meal and SPI (91.1 ± 13.3, *p* = 0.023), as well as between WPI meal and SPI meal (*p* < 0.001, Figure 4B). The C_max_ of several individual carotenoids such as Lut+Zea, Lyc, Pte, Ptf, α-Car, and β-Car were significantly lower (*p* < 0.05) in the presence of SPI, compared to WPI and control meal, except for Lyc and Ptf, where a significant difference was found only between WPI vs. SPI and control vs. SPI, respectively (Figure 4(A2)). Similar as for males, the C_max_ values for TAGs (mg/dL ± SD) did not significantly differ between the three test meals (Figure 4C).

### 3.6. Influence of Proteins on the Time to Peak Plasma TRL-Carotenoid Concentration (T_max_)

The observed time (T_max_) to reach the maximum plasma carotenoid concentration did not change and was found generally at t = 5 h, following the lunch meal. A somewhat bi-phasic absorption pattern was found, with a first peak/plateau at t = 2–3 h and a second one at t = 5 h, which was similar to patterns reported earlier [25]. Of note, at certain times, carotenoid concentrations dropped below baseline, which was attributed to a decreased plasma volume prior to food and fluid intake and an increased circulatory volume after test meal intake.

## 4. Discussion

This study followed our recent in vitro investigation, reporting that proteins may interfere with the bioaccessibility and cellular uptake of carotenoids [14]. The main objective of this clinical trial was to verify these results in vivo. For this purpose, a carotenoid-rich meal was supplemented or not with proteins at a concentration of 50% of the RDA (60 g), and was served to healthy adults on three occasions interspaced by one week, following a randomized cross-over design. The results confirm the previous in vitro observations, as they showed that the two proteins, WPI and SPI, had diverging effects on carotenoid absorption as determined by their AUC of the plasma-TRL fraction. Total carotenoid bioavailability was enhanced by 45% in the presence of WPI, and lowered insignificantly by 8% in the presence of SPI, compared to control meal (no additional proteins).

The tested meals, i.e., a tomato/carrot juice mixture, provided a wide range of carotenoids (Table 3) with concentrations similar to those reported previously [29,30]. At baseline, the participants’ fasting carotenoid plasma concentrations were within the range previously reported for healthy subjects [31]. Following the two-week washout period, the concentrations of total plasma carotenoids were significantly reduced, by approx. 35% at the first clinical visit, with no further significant difference between the baseline concentrations prior to the three clinical visits, suggesting that the washout periods were successful and sufficient to cause a steady reduction of the plasma levels.

The effects of the tested meal, gender, and carotenoid were significant in the applied linear mixed model. The presence of WPI in the tested meal resulted in improved AUC of carotenoids/TAGs over the 10 h course, by 45% compared to the control meal without added proteins, and by 57% compared to SPI-supplemented meal. This is, to our knowledge, the first study finding an effect of proteins on carotenoid absorption in humans.

In our previous in vitro studies, we have reported these two diverging effects of various proteins on the bioaccessibility of carotenoids [14,23,27,32], which strongly depended on the type of protein and carotenoid. In fact, certain proteins with a high surface hydrophobic nature such as SPI, when partially digested, were suspected to aggregate in the aqueous phase and/or adsorb in the form of long polypeptides at the O/W interface [23], thus trapping carotenoids and producing negative interactions, limiting the processing from lipid droplets to mixed micelles [14]. In addition, this would be fostered by the general lower digestibility of SPI vs. WPI [23]. On the other hand, more digestible proteins, such as WPI [23], appeared to foster bioaccessibility and the cellular uptake of carotenoids, via producing peptides of more amphiphilic structure, facilitating the interactions at the surface of the mixed micelles, making them potentially more available for cellular uptake [14]. 

In addition to the effect on total carotenoids, all individual carotenoids were modulated towards higher bioavailability in the presence of WPI. However, these differences were found statistically significant only for Pte, Ptf, α-Car, and β-Car, with strongest responses observed for Pte and Ptf, while plasma Lyc TRL-concentrations did not change between the three tested meals. This is somewhat different from the observations in vitro, where proteins rather fostered the bioavailability of individual carotenes compared to xanthophylls [14]. In those studies, the effect of proteins on the micellization of carotenoids was reflected by their polarity; i.e., while proteins enhanced the bioaccessibility of pure β-carotene, a decrease was observed for lutein, while the influence was somewhat limited regarding lycopene [13,33]. It was postulated that the presence of proteins at the interface may result in a stronger negative interaction of proteins and the more polar xanthophylls, which are preferentially solubilized at the surface of the lipid droplets and later of the mixed micelles [34]. However, more apolar carotenes may also have been bound by the major WPI fraction, the pepsin-resistant β-lactoglobulin, with a high affinity of the carotene to the internal cavity of the β-barrel [35], which could act as a vehicle for hydrophobic compounds with positive effects on carotene solubility. Such interactions may be specific to a certain structure, i.e., the β-ionone cycle and isoprenoid chain, which may explain the more neutral effect of WPI on lycopene bioaccessibility in the present work. Similar results were found in our previous study regarding the effect of proteins on the bioaccessibility of carotenoids from food matrices [14]. However, co-digested proteins fostered the cellular uptake of carotenes and counteracted the negative effect of proteins on xanthophyll bioaccessibility by improving their cellular uptake, especially in the presence of WPI [14], possibly by producing peptides that had more amphiphilic structure and facilitated the interactions of mixed micelles with Caco-2 cells [36]. However, it cannot be excluded that in the present study, the low amount of xanthophylls present in the juice mixture, compared to carotenes, also aided in a more favorable effect on their absorption. 

Interestingly, strongest responses were observed for Pte and Ptf, compared to other carotenoids. These results are in line with our previous findings on bioaccessibility and cellular uptake [14], especially in the presence of WPI. Indeed, in vitro studies have shown that these colorless carotenoids, present in tomato and its products, displayed relatively high bioaccessibility [30], and cellular uptake [37], and are more available than lycopene in healthy adults [38]. This was explained by the higher molecular flexibility and twisting ability [39], allowing a better insertion of these carotenoids into the mixed micelles, leading to a greater micellization efficiency [40].

In addition to the effect of tested meal on the AUC for total carotenoids, AUC values for serum TAGs of WPI-supplemented meal were also significantly higher compared to the control meal, and lower from the SPI meal. It is acknowledged that TAGs are absorbed from the intestine, transported as chylomicrons through the lymphatic system prior to entering the systemic circulation, and are metabolized extravascularly, liberating free fatty acids (FFAs) [41]. During lipid metabolism, FFAs are reincorporated into TAGs in the liver and further transported within lipoproteins. As carotenoid absorption is generally considered an aspect of lipid absorption, it is noteworthy that the lipemic response corresponded with carotenoid absorption [42]. This suggests that absorption of both carotenoids and TAG can be significantly impacted by the presence of proteins, which is further plausible given also that FFA are absorbed from the intestine via mixed micelles. This similarity in absorption patterns was supported by the observed time (T_max_) to reach the maximum plasma carotenoid and serum TAG concentrations (C_max_), which followed in both cases, i.e., TAGs and carotenoids, a biphasic curve pattern, with peaks or at least shoulders at approximately t = 2 h and 5 h postprandially (Figure 3), similar as observed in previous studies [31,43]. Furthermore, C_max_ was significantly altered by protein addition, both for TAGs and for total carotenoids, in a very similar fashion as their corresponding AUCs. 

While the absorption patterns of carotenoids as influenced by WPI and SPI were similar for men and women, the effect of tested meal did not reach significance in women, which could be due to the generally lower responses of AUC in women. In addition, women displayed higher plasma carotenoid concentrations compared to men, possibly proposing a higher carotenoid status in women, which may have resulted in lower fractional uptake of carotenoids from the test meals. The intestine-specific homeobox (ISX), a transcription factor, can reduce absorption of pro-vitamin A carotenoids, given that the body status is high [44]. Previous studies have reported similar findings of higher circulating carotenoids levels in women [45,46,47,48]. While some investigations postulated that this sex difference might be explained by better dietary habits such as higher fruit and vegetable intake in female participants [49,50,51], others have explained the difference by higher levels of visceral white adipose tissue in men and different plasma lipoprotein–lipid profiles. A recent investigation reported that increased body weight and waist circumference were associated with lower total plasma carotenoid concentrations, while elevated plasma LDL-cholesterol and HDL-cholesterol concentrations correlated with higher total carotenoids in plasma, due to elevated transport capacity of carotenoids [52]. This would be in line with the present observations (despite the relatively limited sample size in each gender group), as a higher BMI was observed for men that may have been related to higher adiposity in men in the present study, but higher levels of blood lipids in women at the time of recruitment (Table 1). These aspects may explain, at least in part, that despite the reduction of the postprandial AUC for total carotenoids in female volunteers by approx. 18% for SPI, and an increase of almost 42% for WPI, this influence was statistically not significant compared to the control meal, unlike in males. However, it is worth mentioning that proteins influenced the lipemic response, as the AUC values for TAGs in females were significantly higher for the WPI-supplemented meal vs. the control meal.

When transferring such findings to daily life and practical recommendations, it is recognized, for instance, that salads should be consumed together with a fat-containing salad dressing, to benefit from liposoluble micronutrients and secondary plant compounds bound to the plant matrix [53]. Similar considerations may also apply to promoting food combinations with proteins to maximize carotenoid bioavailability. However, our study has the limitation that we used only proteins added in form of powder, and as our earlier in vitro studies have indicated that the effect for proteins bound within real food matrices such as cod and turkey are somewhat weaker in their interactions with carotenoids [14], more studies in this domain are warranted to examine effects from other protein sources. Adjusting dietary recommendations for subjects with low vitamin A status or low vitamin A intake such as vegetarians, or people living in developing countries where access to meat and thus preformed vitamin A is often limited, could also be considered, i.e., combining carotenoid intake with well digestible proteins, if available. In addition, minor fractions of the population such as those having digestive disturbances and impeded absorption for liposoluble micro-constituents from the diet [32], may also benefit from the knowledge gained from this trial. Finally, the combination of proteins and carotenoids together in food supplements may deserve further attention [54].

## 5. Conclusions

This study shows, for the first time, that proteins added to a test meal rich in carotenoids, at a protein amount equivalent to 50% of the RDA, could improve carotenoid bioavailability in healthy adults. The results suggest that a well digestible protein such as WPI could be beneficial for carotenoid bioavailability, whereas a less digestible one such as SPI may result in hampered availability. It appears that the effect of proteins on the bioavailability of carotenoids, and potentially other liposoluble nutrients [55], depends on the type of protein, but likely also on the individual carotenoid species. Further investigations are necessary to fully understand the complex interactions between proteins and carotenoids.

## Figures and Tables

**Figure 1 antioxidants-10-01748-f001:**
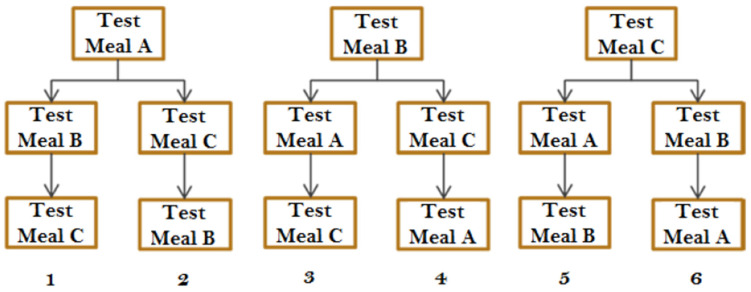
Scheme of breakfast meal sequences for each treatment pattern during the clinical trial phase. Meal A: control, no added protein; Test meal B: added WPI, Test meal C: added SPI. Six final test meal sequences were possible.

**Figure 2 antioxidants-10-01748-f002:**
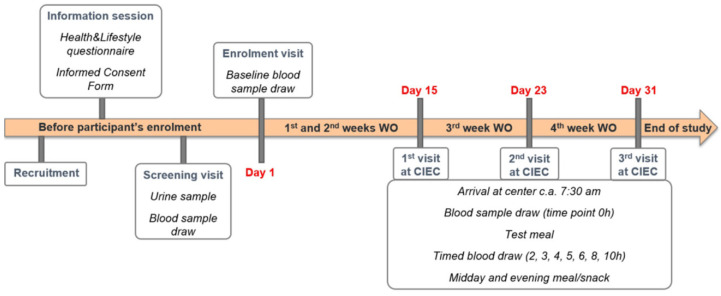
Representative scheme of the participants’ schedule during the trial phase. WO: Washout.

**Figure 3 antioxidants-10-01748-f003:**
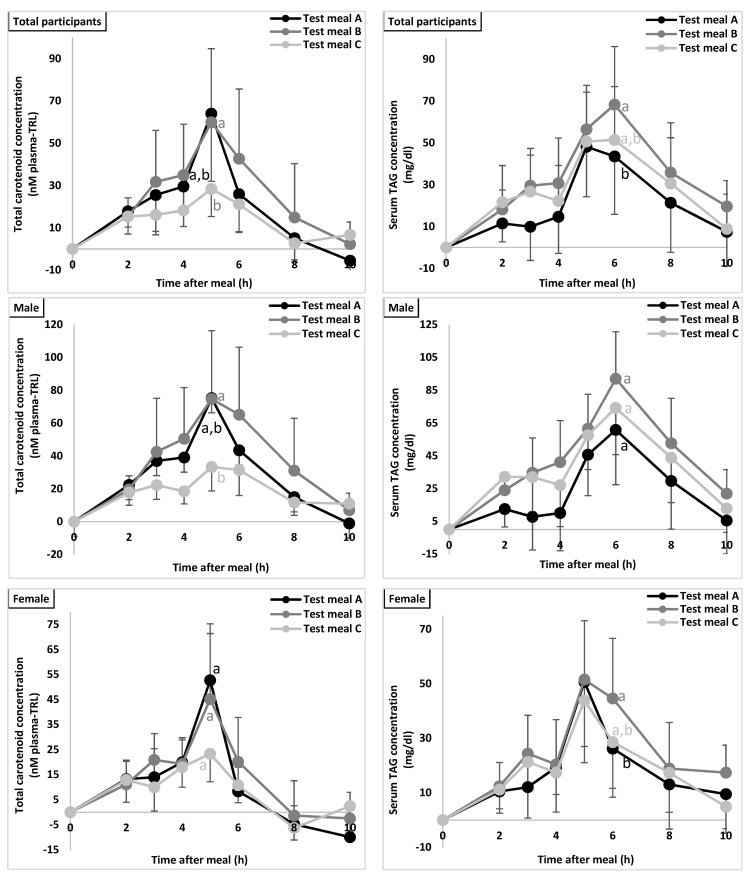
Effect of different protein types on the postprandial serum TAG concentrations and plasma TAG-rich lipoprotein (TRL) concentration of carotenoids. Changes in plasma TRL-carotenoids (left panel) and serum TAGs (right panel) were assessed over a 10 h period after the intake of a test meal without (control, test meal A) or including one of the two investigated proteins, namely whey protein isolate (WPI, test meal B) or soy protein isolate (SPI, test meal C). Plasma TRL-carotenoid concentrations (nM) and serum TAG concentrations (mg/dL) are expressed as mean values (total participants *n* = 24, male/female *n* = 12) with their standard errors. AUC were compared for statistically significant differences as outlined in the “Statistical Analyses” section. Solid lines not sharing the same alphabetical letter (a,b) are considered significantly different.

**Figure 4 antioxidants-10-01748-f004:**
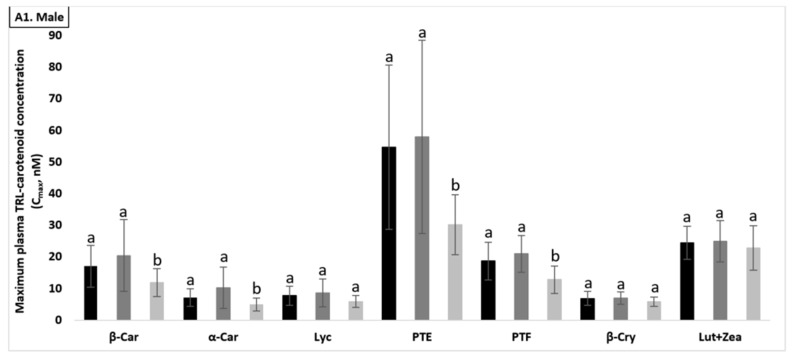
Effect of the tested meals on the maximum concentrations (Cmax) of postprandial plasma TAG-rich lipoprotein (TRL) carotenoids (**A**,**B**) and serum TAGs (**C**). Changes in Cmax of the plasma TRL-carotenoids and serum TAGs that reflect the highest concentrations measured, irrespective of the time point, following the consumption of the ■ control meal, ■ whey protein isolate (WPI-supplemented meal), or ■ soy protein isolate (SPI-supplemented meal). (**A**). individual carotenoid concentration (nM) in the plasma TRL fraction in male (**A1**), female (**A2**), and total participants (**A3**); (**B**). total carotenoid concentration (nM) in the plasma TRL fraction; (**C**). serum TAG concentrations (mg/dL). Concentrations are expressed as mean values (total participants = 24, male/female = 12) with their standard errors. AUC were compared for statistically significant differences as outlined in the “Statistical Analyses” section. Columns labelled without a common superscript (alphabetic letters) differ significantly, *p* < 0.05.

**Table 1 antioxidants-10-01748-t001:** Anthropometric characteristics and fasting blood biochemistry from the twenty-four participants, at the time of recruitment (mean values and standard deviations).

	Male (*n* = 12)			Female (*n* = 12)			All Participants (*n* = 24)		
	Mean (SD)	Max	Min	Mean (SD)	Max	Min	Mean (SD)	Max	Min
Anthropometric measures ^1^									
Age (years)	30.9 (6.4)	46.0	22.0	29.7 (6.3)	42.0	22.0	30.3 (6.3)	46.0	22.0
BMI (kg/m^2^)	25.1 (2.3)	28.7	19.7	22.8 (1.8)	25.7	18.9	23.9 (2.4)	28.7	18.9
Body fat (%)	18.4 (5.7)	27.5	8.9	31.5 (5.0)	38.4	22.9	24.9 (8.5)	38.4	8.9
Waist/hip ratio	0.9 (0.1)	1.0	0.8	0.8 (0.1)	0.9	0.5	0.9 (0.1)	1.0	0.5
Blood biochemistry profile ^2^									
Glucose (mg/dL)	96.6 (8.0)	108.0	82.0	87.9 (7.3)	106.0	79.0	92.3 (8.7)	108.0	79.0
Total cholesterol (mg/dL)	152.3 (31.2)	193.0	94.0	176.4 (36.3)	230.0	99.0	164.3 (35.3)	230.0	94.0
HDL-cholesterol (mg/dL)	48.5 (15.4)	88.0	32.0	67.3 (14.9)	95.0	48.0	57.9 (17.7)	95.0	32.0
Non-HDL-cholesterol (mg/dL)	103.6 (33.7)	161.0	43.0	109.1 (31.6)	152.0	37.0	106.3 (32.1)	161.0	37.0
LDL-cholesterol (mg/dL) ^3^	95.1 (29.3)	144.0	43.0	106.5 (29.7)	143.0	32.0	100.8 (29.4)	144.0	32.0
TAG (mg/dL)	88.7 (44.7)	192.0	44.0	75.4 (29.9)	129.0	39.0	82.0 (37.8)	192.0	39.0

^1^ Measured using impedance method. ^2^ Fasting blood levels. ^3^ Calculated with the Friedewald equation.

**Table 2 antioxidants-10-01748-t002:** Composition of test meals.

	Amounts	Ingredients	Nutritional Values Per Serving
Breakfast meal (mornings)			
Mixture carrot juice/ tomato juice (1:1, *v/v*)	350 mL	99% carrot juice, 1% concentrate from lemon 99.5% tomato juice, 0.5% salt	Energy: 282 kJ/67 kcal/Fat: 0.9 g, of which saturated 0.2 g/Carbohydrates: 13.5 g, of which sugars 13.5 g/Fibers: 1.6 g/Proteins: 2.5 g/Salt: 1.6 g
Peanut oil ^1^	5 mL	Refined peanut oil 100%	Energy: 185 kJ/45 kcal/Fat: 45 g, of which saturated: 0.85 g, mono-unsaturated: 3.3 g, polyunsaturated: 0.85 g
Proteins ^1^			
Control (A)	0 g	n.a. ^3^	n.a. ^3^
WPI (B)	30 g	Whey protein isolate, emulsifier (soy lecithin)	Energy: 503 kJ/119 kcal/Fat: 0.3 g, of which saturated 0.2 g/Carbohydrates: 0.5 g, of which sugars 0.5 g/ Fibers: 1 g/Proteins: 28.0 g/Salt: 0.2 g
SPI (C)	30 g	Soy protein isolate, emulsifier (E322-sunflower)	Energy: 472 kJ/113 kcal/Fat: 0.3 g, of which saturated 0.06 g/Carbohydrates: 0.3 g, of which sugars 0.03 g/Fibers: 0.3 g/Proteins: 27.2 g/Salt: 0.09 g
Toasted bread	40 g	Wheat flour (65%), water, sugar, colza oil, salt, vinegar, yeast, bean flour, wheat gluten, aroma (contains alcohol), acerola extract	Energy: 457 kJ/108 kcal/Fat: 1.6 g, of which saturated 0.2 g/Carbohydrates: 19.6 g, of which sugars 3.0 g/Fibers: 1.6 g/Proteins: 3.0 g/Salt: 0.4 g
Butter ^2^	12.5 g	Fat 82%	Energy: 385 kJ/94 kcal/Fat: 10.3 g, of which saturated 7.1 g/Carbohydrates: 0.1 g, of which sugars 0.1 g/Proteins: 0.1 g/Salt: <0.1 g
Cream cheese ^2^	19 g	Rehydrated skimmed milk, cheese, butter, milk mineral concentrate.	Energy: 172 kJ/41 kcal/Fat: 3.3 g, of which saturated 2.2 g/Carbohydrates: 0.9 g, of which sugars 0.9 g/Proteins: 2.0 g/Salt: 0.32 g
Water	300 mL	Analysis (mg/L): Ca:5; Mg:2; Na:3; K:0.5; Cl:5; SO_4_:4; NO_3_:1.5; HCO_3_:17; SiO_2_:7	n.a. ^3^
Total energy Control meal WPI meal SPI meal			1281 kJ/355 kcal 1759 kJ/467 kcal 1753 kJ/468 kcal
Lunch (4 h after breakfast meal)			
Toasted bread	60 g		Energy: 686 kJ/163 kcal/Fat: 2.5 g, of which saturated 0.3 g/Carbohydrates: 29.3 g, of which sugars 4.6 g/Fibers: 2.3 g/Proteins: 4.6 g/ Salt: 0.7 g
Butter ^2^	12.5 g	Fat 82%	Energy: 385 kJ/94 kcal/Fat: 10.3 g, of which saturated 7.1 g/Carbohydrates: 0.1 g, of which sugars 0.1 g/Proteins: 0.1 g/Salt: <0.1 g
Turkey ^2^	60 g	Turkey fillet 90%, salt, dextrose, spices, acidity regulator (sodium lactate), stabilizers (sodium citrate, carrageenans), flavor, antioxidant (sodium ascorbate), preservative (sodium nitrite)	Energy: 247 kJ/58 kcal/Fat: 0.9 g, of which saturated 0.3 g/Carbohydrates: 0.5 g, of which sugars 0.4 g/Proteins: 12.0 g/Salt: 1.2 g
Greek yogurt	150 g	Pasteurized cow milk, milk cream, milk protein, yogurt culture	Energy: 804 kJ/194 kcal/Fat: 15 g, of which saturated 11.3 g/Carbohydrates: 5.3 g, of which sugars 5.3 g/Fibers: <0.5 g/Proteins: 8.3 g/ Salt: 0.2 g
Apple	150 g	n.a. ^3^	Energy: 321 kJ/77 kcal/Fat: >0.5 g, of which saturated >0.1g/Carbohydrates: 16.5 g, of which sugars 15.0 g /Fibers: 3.8 g/Proteins: <0.5 g/Salt: <0.01 g.
Water	300 mL	Analysis (mg/L): Ca:5; Mg:2; Na:3; K:0.5; Cl:5; SO_4_:4; NO_3_:1.5; HCO_3_:17; SiO_2_:7	n.a. ^3^
Total energy			2443 kJ/586 kcal

^1^ Dissolved/added to carrot and tomato mixture. ^2^ Spread on the toasted bread. ^3^ Not applicable.

**Table 3 antioxidants-10-01748-t003:** Carotenoid contents (mg/100 g) of the tomato and carrot juices ^1^, as determined by HPLC.

Breakfast Meal	Lut+Zea	Ptf	Pte	α-Car	β-Car	Lyc	Total
Tomato juice	0.04	1.15	1.63	0.02	0.25	7.79	10.9
Carrot juice	0.09	0.88	0.49	1.70	4.40	0.04	7.6
Juice mix 100 g	0.06	1.01	1.06	0.86	2.32	3.91	9.25
Juice mix 350 g	0.2	3.5	3.7	3.0	8.1	13.7	32.4

^1^ Each value represents the mean of *n* ≥ 3 replicates. Lut+Zea: lutein+zeaxanthin; Ptf: phytofluene; Pte: phytoene; α-Car: α-carotene; β-Car: β-carotene; Lyc: lycopene.

**Table 4 antioxidants-10-01748-t004:** Blood plasma carotenoid concentrations (nM) * of the participants at the baseline (D_0_), first clinical visit (V_1_), second clinical visit (V_2_), and third clinical visit (V_3_).

Gender	Carotenoids/Visits	Lut+Zea	β-Car	α-Car	Lyc	Pte	Ptf	β-Cry	Total carot. ^†^
Female (*n* = 12)	D_0_	282.4 (±237.6) ^a^	85.3 (±76.0) ^a^	12.9 (±8.3) ^a^	53.3 (±39.5) ^a^	31.7 (±14.2) ^a^	60.0 (±46.1) ^a^	56.0 (±35.9) ^a^	581.3 (±361.8) ^a^
	V_1_	214.4 (±202.8) ^b^	55.4 (±61.6) ^b^	10.0 (±5.6) ^a^	42.4 (±52.0) ^b^	22.6 (±35.6) ^b^	44.9 (±45.5) ^b^	39.5 (±23.1) ^b^	429.3 (±386.1) ^b^
	V_2_	173.6 (±134.9) ^b^	55.2 (±50.7) ^b^	10.8 (±4.8) ^a^	40.2 (±46.1) ^b^	17.8 (±11.9) ^b^	44.4 (±39.8) ^b^	34.8 (±20.7) ^b^	376.9 (±271.6) ^b^
	V_3_	175.0 (±120.4) ^b^	58.0 (±57.7) ^b^	11.2 (±5.0) ^a^	35.3 (±35.3) ^b^	15.8 (±9.0) ^b^	46.5 (±46.8) ^b^	35.4 (±22.8) ^b^	377.3 (±262.1) ^b^
Male (*n* = 12)	D_0_	100.9 (±59.8) ^a^	63.5 (±59.6) ^a^	22.5 (±24.3) ^a^	40.8 (±26.9) ^a^	38.7 (±30.4) ^a^	29.4 (±21.0) ^a^	32.9 (±16.7) ^a^	328.8 (±139.3) ^a^
	V_1_	61.6 (±37.9) ^a^	38.0 (±34.0) ^b^	13.5 (±13.9) ^b^	23.1 (±15.4) ^b^	11.6 (±6.0) ^b^	14.2 (±7.4) ^b^	19.6 (±10.0) ^a^	181.6 (±74.0) ^b^
	V_2_	79.8 (±61.3) ^a^	37.0 (±30.1) ^b^	13.0 (±11.9) ^b^	19.3 (±11.8) ^b^	10.8 (±6.7) ^b^	13.8 (±7.0) ^b^	19.9 (±9.5) ^a^	193.7 (±71.4) ^b^
	V_3_	79.0 (±62.8) ^a^	35.5 (±29.2) ^b^	13.2 (±11.7) ^b^	18.0 (±10.9) ^b^	12.2 (±6.5) ^b^	14.5 (±9.7) ^b^	21.1 (±13.3) ^a^	193.7 (±76.9) ^b^
TotalParticipants (*n* = 24)	D_0_	191.7 (±193.1) ^a^	74.4 (±67.7) ^a^	17.7 (±18.4) ^a^	47.0 (±33.6) ^a^	35.2 (±23.5) ^a^	44.5 (±38.3) ^a^	44.7 (±29.8) ^a^	455.0 (±297.5) ^a^
	V_1_	138.0 (±162.6) ^b^	46.7 (±49.5) ^b^	11.7 (±10.5) ^b^	32.7 (±38.8) ^b^	17.1 (±25.6) ^b^	29.5 (±35.5) ^b^	29.6 (±20.2) ^b^	305.4 (±299.9) ^b^
	V_2_	126.7 (±113.1) ^b^	46.1 (±41.9) ^b^	11.9 (±8.9) ^b^	29.8 (±34.6) ^b^	14.3 (±10.1) ^b^	29.1 (±32.0) ^b^	27.4 (±17.5) ^b^	285.3 (±215.6) ^b^
	V_3_	127.0 (±105.9) ^b^	46.8 (±46.2) ^b^	12.2 (±8.9) ^b^	26.7 (±30.3) ^b^	14.0 (±8.0) ^b^	30.5 (±36.9) ^b^	28.3 (±28.3) ^b^	285.5 (±210.9) ^b^

* Each value represents the mean of *n* ≥ 3 replicates. Lut+Zea: lutein+zeaxanthin; β-Cry: β-cryptoxanthin; Ptf: phytofluene; Pte: phytoene; α-Car: α-carotene; β-Car: β-carotene; Lyc: lycopene. Values in the same column of the same group with a different superscript letter differ significantly. **^†^** sum of all listed carotenoids.

**Table 5 antioxidants-10-01748-t005:** Effect of the test meals on the postprandial plasma AUC of TAG-rich lipoprotein (TRL) concentration of carotenoids and TAGs in serum.

Carotenoids ^†^ /TAGs	Breakfast Meal	Female	Male	Total Participants
Lut + Zea	Control	−16.4 ± 4.5 ^a^	15.0 ± 2.4 ^a^	−0.6 ± 1.8 ^a^
	WPI	−7.1 ± 4.5 ^a^	17.5 ± 2.4 ^a^	5.2 ± 1.8 ^a^
	SPI	0.2 ± 4.5 ^a^	28.7 ± 2.4 ^a^	14.4 ± 1.8 ^a^
β-Car	Control	1.7 ± 3.1 ^a^	24.9 ± 3.2 ^a,b^	13.3 ± 1.4 ^a,b^
	WPI	1.7 ± 3.1 ^a^	43.4 ± 3.2 ^b^	22.6 ± 1.4 ^b^
	SPI	−0.2 ± 3.1 ^a^	6.6 ± 3.2 ^a^	3.2 ± 1.4 ^a^
α-Car	Control	4.2 ± 0.4 ^a,b^	10.7 ± 1.6 ^a,b^	7.4 ± 0.6 ^a,b^
	WPI	5.9 ± 0.4 ^b^	22.2 ± 1.6 ^b^	14.1 ± 0.6 ^b^
	SPI	2.0 ± 0.4 ^a^	4.2 ± 1.6 ^a^	3.1 ± 0.6 ^a^
Lyc	Control	−8.8 ± 1.7 ^a^	1.7 ± 1.2 ^a^	−3.6 ± 0.7 ^a^
	WPI	−13.5 ± 1.7 ^a^	6.4 ± 1.2 ^a^	−3.6 ± 0.7 ^a^
	SPI	−8.3 ± 1.7 ^a^	4.2 ± 1.2 ^a^	−2.0 ± 0.7 ^a^
Pte	Control	94.5 ± 2.3 ^b^	176.9 ± 9.1 ^a,b^	135.7 ± 3.3 ^b^
	WPI	112.1 ± 2.3 ^b^	207.6 ± 9.1 ^b^	159.9 ± 3.3 ^b^
	SPI	63.3 ± 2.3 ^a^	100.5 ± 9.1 ^a^	81.9 ± 3.3 ^a^
Ptf	Control	28.9 ± 1.7 ^a^	49.2 ± 2.1 ^a^	39.0 ± 0.9 ^a^
	WPI	40.1 ± 1.7 ^a^	69.2 ± 2.1 ^b^	54.7 ± 0.9 ^b^
	SPI	26.3 ± 1.7 ^a^	36.2 ± 2.1 ^a^	31.2 ± 0.9 ^a^
β-Cry	Control	−10.2 ± 0.7 ^a^	0.7 ± 0.8 ^a^	−4.7 ±0.7 ^a^
	WPI	−6.2 ± 0.7 ^a^	−1.5 ± 0.8 ^a^	−3.8 ± 0.3 ^a^
	SPI	−6.5 ± 0.7 ^a^	2.0 ± 0.8 ^a^	−2.3 ± 0.7 ^a^
Totalcarot. ^‡^	Control	93.8 ± 9.7 ^a^	279.2 ± 17.4 ^a,b^	186.5 ± 7.0 ^a,b^
	WPI	133.1 ± 9.7 ^a^	364.5 ± 17.4 ^b^	248.8 ± 7.0 ^b^
	SPI	76.8 ± 9.7 ^a^	171.5 ± 17.4 ^a^	124.1 ± 7.0 ^a^
TAGs	Control	165.3 ± 6.3 ^a^	232.1 ± 25.8 ^a^	196.3 ± 9.6 ^a^
	WPI	238.2 ± 6.3 ^b^	426.7 ± 25.8 ^a^	332.4 ± 9.3 ^b^
	SPI	191.5 ± 6.3 ^a,b^	360.8 ± 25.8 ^a^	276.2 ± 9.3 ^a,b^

Values represent mean ± SEM. Means without a common superscript letter differ significantly, *p* < 0.05. Lut+Zea: lutein+zeaxanthin; β-Cry: β-cryptoxanthin; Ptf: phytofluene; Pte: phytoene; α-Car: α-carotene; β-Car: β-carotene; Lyc: lycopene. Total carot.: total carotenoids. **^†^** AUC results for carotenoids in the plasma TAG-rich lipoprotein (TRL, nmol × h/L) fractions, and for serum TAGs (mg/dL × h/L), following the consumption of a test meal without added protein (control meal), or with either whey protein isolate (WPI-supplemented meal) or soy protein isolate (SPI-supplemented meal) in male (*n* = 12), female (*n* = 12) or in total participants (*n* = 24). **^‡^** Sum of all individual carotenoids.

## Data Availability

The data presented in this study are available in article. All data necessary for statistical analysis have been gathered by means of a specifically designed electronic case report form (eCRF) for each patient (Ennov Clinical, Ennov Group, Paris). A clearly designated clinical research nurse from CIEC was responsible for its completion. A clinical research associate from CIEC conducted quality control on the data reported in eCRF.

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
