# Peer review of "Whey- and Soy Protein Isolates Added to a Carrot-Tomato Juice Alter Carotenoid Bioavailability in Healthy Adults"

_antioxidants, 2021, doi:10.3390/antiox10111748_

Round 1
Reviewer 1 Report
The study on how the presence and the nature of proteins in a diet can interact with carotenoid bioavailability is original and the experimental design seems to me well performed. The document is well written and elements are clearly presented most of the time. Besides some minor recommendations, my major points are :
- To verify the eligibility of the statistical analysis performed because the use of the type of carotenoid as a fixed effect in the model is unexpected. Moreover, if eligible, the results are rather not presented and used (no statistical comparisons between the carotenoid concentrations, AUC, Cmax and Tmax). Additionally, carotenoid concentrations were also used as a dependent variable so it is confusing. It can be also notice that some superscripts are questioning in table 4 (for example V1 values for carotenes in males are noted b but it seems strange) and 5 where for several variables, mean values were noted a / a,b / a,c so it would mean that they are not different having a in common so they could be all noted “a”, isn’t it (also in figure 4)?
- I didn’t understand the results of “all carotenoids/TAG investigated (statistically pooled) combined”. These results would reflect the values for bioavailabilities but it is not clear how they were obtained and they are not presented in tables or figures contrary to others (lines 358-361).
Minor revisions in the additional file

Author Response
The study on how the presence and the nature of proteins in a diet can interact with carotenoid bioavailability is original and the experimental design seems to me well performed. The document is well written and elements are clearly presented most of the time.
Reply: We appreciate the overall assessment and remarks of the referee.
Besides some minor recommendations, my major points are:
To verify the eligibility of the statistical analysis performed because the use of the type of carotenoid as a fixed effect in the model is unexpected.
Reply: The type of carotenoids was chosen as a fixed effect in order to investigate whether the observed outcome (e.g. AUC or Cmax) depended on the type of carotenoid, which surely is an important question. This was the most “complete” model possible, which also avoided merely running repeated models with each carotenoid as an endpoint (risking wrong positives by just having too many models). In addition, the same model was employed in our earlier publication which followed a very similar design (1), and the statistical approach was verified by our statistical unit (CCMS at DOPH).
Moreover, if eligible, the results are rather not presented and used (no statistical comparisons between the carotenoid concentrations, AUC, Cmax and Tmax).
Reply: The main aim was to compare the effect of proteins on the total and individual carotenoids, and not necessarily between the carotenoids, as we assumed that there surely would exist differences between their absolute amount absorbed. Since carotenoids are present in quite varying amounts in the matrix already, they will be absorbed (again speaking in absolute amounts, AUC etc.) to a different extent. The same was true for Cmax and Tmax. Thus, we studied whether carotenoids were differently effected by the addition of proteins, but did not find it meaningful to compare their absolute AUCs, Cmax and similar.
Additionally, carotenoid concentrations were also used as a dependent variable so it is confusing.
Reply: It is correct that carotenoid concentrations in the circulatory system, expressed as AUC of plasma-triacyl-rich-lipoprotein fractions, were the main observed outcome (dependent) variable. This was the main targeted endpoint. We have removed the line that carotenoid type was also used as a dependent variable, which was incorrectly stated. Carotenoid type, not any carotenoid concentration, on the other hand, was inserted into the model as a fixed factor. Again, we repeated the statistics we applied from earlier human studies (1). We have further tried to improve the description of the statistical part.
It can be also notice that some superscripts are questioning in table 4 (for example V1 values for carotenes in males are noted b but it seems strange), and 5 where for several variables, mean values were noted a / a,b / a,c so it would mean that they are not different having a in common so they could be all noted “a”, isn’t it (also in figure 4)?
Reply: We apologize for these errors; the superscripts are now corrected in Tables 4 and 5, as well as in Figures 3 and 4.
I didn’t understand the results of “all carotenoids/TAG investigated (statistically pooled) combined”. These results would reflect the values for bioavailabilities but it is not clear how they were obtained and they are not presented in tables or figures contrary to others (lines 358-361).
Reply: These results were obtained by considering all carotenoid types and TAGs together, i.e. statistically pooled. This is the first outcome from the global linear mixed model, considering that type of carotenoids and also TGs were included in the model within the fixed factor “type of carotenoids”. We think that such results are still interesting to report, but not necessarily to be presented in tables or figures. Instead, it is more appropriate to present the results obtained from individual concentrations for each carotenoid type and TAG as done in Table 5.
I have only minor considerations:
o avoid the “/”that could be interpreted several ways like Lut/zea (= lut+zea; by the way, how are they quantified ?), serum TAG/plasma TRL, carotenoids/TAG…
Reply: We agree with the referee, we replaced “Lut/Zea” by “Lut+Zea” throughout the text. For the results represented in the Table 5 (bioavailability), they were both quantified in the plasma AUC TRL fraction (ref. Table 5), while in Table 4, the values represent the blood plasma concentrations of circulating Lut and Zea. We have chosen to combine the two carotenoids in order to avoid possible errors in quantification, due to the two slightly overlapping elution profile (HPLC) and similar spectra of Lut and Zea. It should be noted that the majority was Lut and only a small amount of Zea was detected, and we thought that this was also a good reason to combine the two carotenoids. Regrettably, this is not an uncommon limitation to the HPLC-based quantification of carotenoids and is done in many publications dealing with carotenoids.
please choose between meal A, B or C and Control, WPI, SPI (preferably the latter)
Reply: We thank the referee for the suggestion. We have replaced the meal A, B or C by Control, WPI, SPI throughout the text. However, we think that is superior to keep test meal A, B and C at least in some tables/figures, as they were used for example to build the different sequences for the trial and text space in figures/tables is more limited. Additional details are now added throughout the text when it was needed to clarify that test meal A referred to control meal, test meal B to WPI-supplemented meal, and test mea C to SPI-supplemented meal, in order to avoid any misunderstanding.
Line 122: one “(“ to remove
Reply: Thanks, it is now removed.
Line 133: only vegan? what about vegetarian
Reply: We agree with the referee, vegetarian subjects were also excluded.
Figure 1 : not necessary because it was well presented in lines 134-136
Reply: We agree that the information in the figure 1 is partly replicated by the text, however, we feel that it could help the reader to better understand the text and aids in clarity, as we think that not everyone is familiar with sequence building including the randomized block design in human trials.
Line 165-166: the final inclusion should be given at the end of the selection procedure
Reply: We thank the referee for the comment; the final inclusion was moved as suggested (please see line 199).
Line 188: maybe you could precise that clinical visits were days when test meals were given
Reply: Thank you. It is now specified that depending on the test day, proteins (either SPI or a WPI) were or were not added to the breakfast meal (please see line 183).
Line 268: what was the role of the internal standard here? As I understand, you added it in samples after extraction. It would have been more interesting to add it in plasma before extraction to take into account the losses during the process.
Reply: In this case, the major objective of the internal standard (IS) was to assure chromatographic consistency, i.e. monitoring that the injection needle was not blocked, rather than constituting an internal standard for the whole procedure. We have been using this method for some time, and have tested the reliability (e.g. by spiking experiments) from plasma earlier, and with our extraction method losses appeared to be very low and reproducible.
Line 335: how were performed the triplicates: one at each clinical visit or analytical replicates at a same moment?
Reply: Three independent samples were extracted at one same time from each of the tomato juice and carrot juice lots that were used in the trial, i.e. these were independent replicates, not technical replicates. As the same badges of juice were given to the same subjects, differences in carotenoid concentrations would not have played a role regarding the influence of proteins, and we could not detect any noticeable difference over the time of the study in carotenoid composition of the juices.
Line 344 : I don’t agree with that for a-carotene in women, Lut/zea and b-crypto for men
Reply: We agree with the referee. We have mentioned the exceptions found in men (see line 334), while this was corrected for women (ref. Table 4).
Line 410: inversion between right and left panel
Reply: We thank the referee for the comment. It is now corrected (see line 410)
From Line 419: when significant differences were observed, it could be interesting to give the % of variation rather than only lower or higher
Reply: The percentages are now added (please see line 419).
Line 449: more 5 hours than 4 according to figures
Reply: We agree with the referee. It is now corrected.
Lines 463-465: reference to lines 359-360? not so clear that these values are those of the carotenoid bioavailabilities
Reply: We apologize for this confusion. It is now specified that the values referred to AUC of carotenoids/TAGs combined, i.e. all measured outcomes combined in the statistical model.
Lines 575-579: for some aspects like here, the discussion part could be modulated because the experimental design of the study was limited in term of participants.
Reply: We thank the referee for this comment. The limited sample size for each gender group is now mentioned (see line 562).

Reviewer 2 Report
This is an excellent research paper that sheds light on obscure points on the bioavailability of dietary carotenoids. The design, methodology and discussion are appropriate and the main findings are of great interest. The quality of the study is unarguable. I do not have criticisms.
Author Response
Reply: We appreciate your kind assessment of the manuscript.

This manuscript is a resubmission of an earlier submission. The following is a list of the peer review reports and author responses from that submission.